

# Spatiotemporal dynamics of habitat suitability for the Ethiopian staple crop, *Eragrostis tef* (teff), under changing climate

Dinka Zewudie[1], Wenguang Ding[1], Zhanlei Rong[2], Chuanyan Zhao[3] and Yapeng Chang[3]

[1] College of Earth and Environmental Sciences, Lanzhou University, Lanzhou, China
[2] College of Geographical Science, Qinghai Normal University, Xining, China
[3] State Key Laboratory of Grassland Agro-ecosystems, College of Pastoral Agriculture Science and Technology, Lanzhou University, Lanzhou, China

Corresponding author
Wenguang Ding, wgding@lzu.edu.cn

## ABSTRACT

Teff (*Eragrostis tef* (Zucc.) Trotter) is a staple, ancient food crop in Ethiopia. Its growth is affected by climate change, so it is essential to understand climatic effects on its habitat suitability in order to design countermeasures to ensure food security. Based on the four Representative Concentration Pathway emission scenarios (i.e., RCP2.6, RCP4.5, RCP6.0 and RCP8.5) set by the Intergovernmental Panel on Climate Change (IPCC), we predicted the potential distribution of teff under current and future scenarios using a maximum entropy model (Maxent). Eleven variables were selected out of 19, according to correlation analysis combined with their contribution rates to the distribution. Simulated accuracy results validated by the area under the curve (AUC) had strong predictability with values of 0.83–0.85 for current and RCP scenarios. Our results demonstrated that mean temperature in the coldest season, precipitation seasonality, precipitation in the cold season and slope are the dominant factors driving potential teff distribution. Proportions of suitable teff area, relative to the total study area were 58% in current climate condition, 58.8% in RCP2.6, 57.6% in RCP4.5, 59.2% in RCP6.0, and 57.4% in RCP8.5, respectively. We found that warmer conditions are correlated with decreased land suitability. As expected, bioclimatic variables related to temperature and precipitation were the best predictors for teff suitability. Additionally, there were geographic shifts in land suitability, which need to be accounted for when assessing overall susceptibility to climate change. The ability to adapt to climate change will be critical for Ethiopia's agricultural strategy and food security. A robust climate model is necessary for developing primary adaptive strategies and policy to minimize the harmful impact of climate change on teff.

## INTRODUCTION

Climate is the key controlling factor in the distribution of species, and variations in habitat distribution patterns can be attributed to climate change (*Parmesan & Yohe, 2003*; *Lenoir*

*et al., 2008*; *Bertrand et al., 2011*; *Guo et al., 2016*). The global average surface temperature has caused warming of 0.8 °C over the past century (*Hansen et al., 2006*; *Yumbya et al., 2014*), and is accelerating, with a 0.6 °C increase in the last four decades (*Hansen et al., 2010*; *Suwannatrai et al., 2017*). The International Panel on Climate Change (*IPCC, 2007*) reports that over the course of this century, net carbon uptake by the terrestrial ecosystem made a peak mid-century and then weakened or even reversed the increase on climate change. Global average temperature increases exceeding the 1.5–2.5 °C range, and the concomitant $CO_2$ concentration is projected to trigger major changes in the ecosystem structure and function, alter species ecological interactions and produce shifts in species' geographical ranges with predominantly negative consequences for biodiversity and ecosystem goods and services, e.g., water and food supply (*Kumar, 2012*; *Liao & Chang, 2014*; *Pachauri et al., 2014*). Evidence suggests that a significant impact of global warming is already discernible in animal and plant populations (*Root et al., 2003*).

Temperature in Africa is expected to increase faster than the global average throughout the 21st century (*Christensen & Christensen, 2007*; *Joshi et al., 2011*; *James & Washington, 2013*. However, future values for Africa estimated by the *Intergovernmental Panel on Climate Change (2007)* are relatively uncertain (*Solomon et al., 2007*). In particular, Ethiopia is predicted to be highly vulnerable to climate change, manifesting an increased warming accompanied by inconsistent rainfall leading to decreased crop security (*Conway & Schipper, 2011*).

Climate change may increase the risk for food production in many African countries as their agricultural systems are mostly rain-fed. These impacts on livelihood, will undoubtedly mean that countries will have to change their farming policies (*Dinar, 2007*). In Ethiopia, agriculture plays a crucial role in enhancing economic growth and contributes up to half of the Gross Domestic Product (GDP) (*Gebrehiwot & Van Der Veen, 2013*). Despite being one of the largest cereal producers in East Africa, the country is still not self-sufficient; 85% of Ethiopians live in agrarian areas relying on agricultural production as a means of living (*Gebrehiwot & Van Der Veen, 2013*). Moreover, many rural Ethiopians face long or short seasonal droughts, leading to crop failure (*Tefera & Quintin, 2012*; *Kamali et al., 2018*). "Green famines" are caused in Ethiopia by a seasonal pattern shift with delayed rains for several weeks or starting and stopping suddenly in critical germination periods. Consequently, crops are lost while the natural vegetation is able to resume normal phenological cycles, thus providing a green landscape (*Evangelista, Young & Burnett, 2013*). Changing patterns of drought and precipitation in Ethiopia have already been documented by other researchers, and greater changes are expected under future conditions (*Deressa & Hassan, 2009*; *Viste, Korecha & Sorteberg, 2013*). Cereal crops face an especially critical threat due to climate change (*Ledig et al., 2010*), affecting human populations that rely on only a few edible species for nutrition and sustenance out of more than 50,000 known edible species available worldwide (*Cheng et al., 2017*).

Teff (*Eragrostis tef* (Zucc.) Trotter), is an ancient and major staple food crop in Ethiopia, accounting for the largest share of land cultivation for cereal crops, 24% nationwide (*Taffesse, Dorosh & Gemessa, 2012*), showing a wide geographic condition and marked

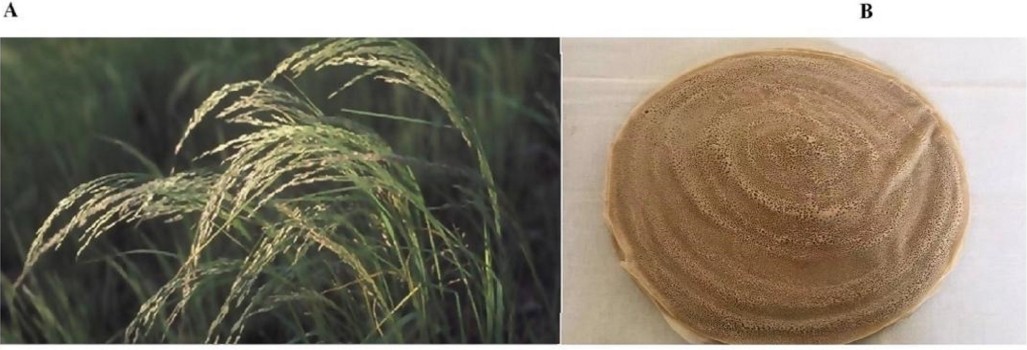

Photo by Gamechis E. Hambisa

**Figure 1** **(A)** The crop teff, a fine-grain annual cereal (Source: FAO http://www.fao.org/traditional-crops/teff/en/); **(B)** injera made from teff is a staple food product in Ethiopia.

plasticity by growing in 10 to 18 agro ecological zones (*Assefa et al., 2001*). Teff is a warm-season annual crop that produces very small grains and is also gluten-free and high in iron (Fig. 1A) (*Rosell et al., 2014*). Teff contains a huge amount of nutrients, and moreover is tolerant of extremes such as drought, Water logging, pests, and diseases (*Cheng et al., 2017*).

Teff crops occupy nearly 30,000 km$^2$ (*CSA, 2014*) about 30% of the total area covered by cereals in all administrative regions, predominantly on the highlands of Tigray, Amhara, Oromia, and SNNPR regions (*Wondimu & Tekabe, 2001*). Teff growth has been reported at altitudes up to 3,000 m.a.s.l. ranging from semiarid areas with low rainfall to areas with high rainfall. Under current conditions, teff is suitable in most areas of the country covering diverse agro ecological regions where millions of people rely on it as a daily food–indeed, nearly 60% of the Ethiopian population.

However, due to climate change, it is one of the most vulnerable crops in the region (*Evangelista, Young & Burnett, 2013*). It is estimated that enjera, made from teff, provides up to two-thirds of the food consumed by Ethiopians (Fig. 1B) (*Cheng et al., 2017*; *VanBuren et al., 2020*). Therefore, predicting potential teff distribution is necessary to take safe and effective countermeasures to reduce its ecological risk under climate change (*Kamilar & Beaudrot, 2013*).

Maxent is an ecological tool and statistical model used to examine ecological processes and interactions across spatial and temporal scales (*Elith et al., 2006*; *Phillips, Anderson & Schapire, 2006*) that has been employed to explore a range of species, habitats and ecosystem conditions (*Alberto et al., 2013*; *Brown, 2014*; *Beltramino et al., 2015*). Species distribution models (SDMs) are a subset of the approaches outlined and were developed by combining current and historical species distribution data with relevant environmental variables to explain both occurrence and abundance of organisms in ecosystems (*Peterson, 2006*; *Zimmermann et al., 2010*; *Caminade et al., 2012*; *Kamilar & Beaudrot, 2013*; *Guo et al., 2017b*). A variety of SDMs have been developed to predict species distributions under different climate scenarios (e.g., RCP). Commonly used SDM models including the

Genetic Algorithm for Rule-set Production (GARP), BIOCLIM and Ecological Niche Factor Analysis (ENFA) have proved to be essential for predicting target species distributions under current, and also future climate scenarios (*Rong et al., 2019*; *Tognelli et al., 2009*). Maxent (maximum entropy) has been widely used due to its many advantages, including its ability to: deal with incomplete data, small sample size, species presence data, both continuous and categorical environmental data; reduce laborious jobs in data collection; facilitate model interpretation, and for its prediction accuracy and reliability (*Guo et al., 2017a*; *Rong et al., 2019*). SDMs assume that available presence locations represent a random (representative) sample in the geographical space, with no spatial dependencies and bolstered by data-rich technologies, such as geographical information systems (GIS) (*Rushton, Ormerod & Kerby, 2004*; *Elith et al., 2011*).

Teff had been an important crop in the past and will continue to be a staple food for the majority of the Ethiopian people in the future. Being a major staple crop, teff is an essential food in Ethiopia, so any impact on suitable areas for its cultivation due to climate change will have a direct impact on food security. Unfortunately, there is already a huge deficit between teff production and the national demand, which will be exacerbated by the effects of climate change. Therefore, as long as teff production decreases against a rising demand due to Ethiopia's fast-growing population, the teff price will increase. As cultural food insecurity remains widespread in Ethiopia, policy makers should pay attention to areas with low habitat suitability and resilience for teff. It is important to maintain nutrition and sustainable food security for the Ethiopian people with the implementation of a strategy to foster diversification and avoid over reliance on this crop.

In a climate change context, understanding and planning for crop resilience is crucial for protection of global food supplies, and therefore research on key crops is needed for decision makers to plan and strategize future actions (*Cowie et al., 2018*). In line with this, the major objectives of this study are to: (1) predict the potential distribution of teff species under current climates; (2) forecast suitable areas for teff species under four future climate scenarios, and (3) evaluate the effects of climate change on teff distribution. Identifying shifts in the ranges of suitable areas under future scenarios is a novel approach in the study and we also shown the full workflow on which every step of analyses of our work is delineated in Fig. 2.

## MATERIALS AND METHODS

In this study, we analyzed the geographic distribution of *Eragrostis tef* occurrence from Ethiopian Biodiversity Institute (EBI) (https://www.ebi.gov.et/) and Ethiopian National Herbarium, Addis Ababa University with Maxent climate envelope model (CEM) under four climate change scenarios.

### Study area

Ethiopia lies in northeast Africa, and is located between latitude 3.30°–15°N and longitude 33°–48°E covering $1.13 \times 10^6$ km$^2$ with heterogenous landscapes including mountains, hills, and flat regions with elevations from below sea level to more than 4,000 m.a.s.l. (*Tilamun & Emily, 2012*) (Fig. 3). Due to this varied relief, Ethiopia has a variety of climates going
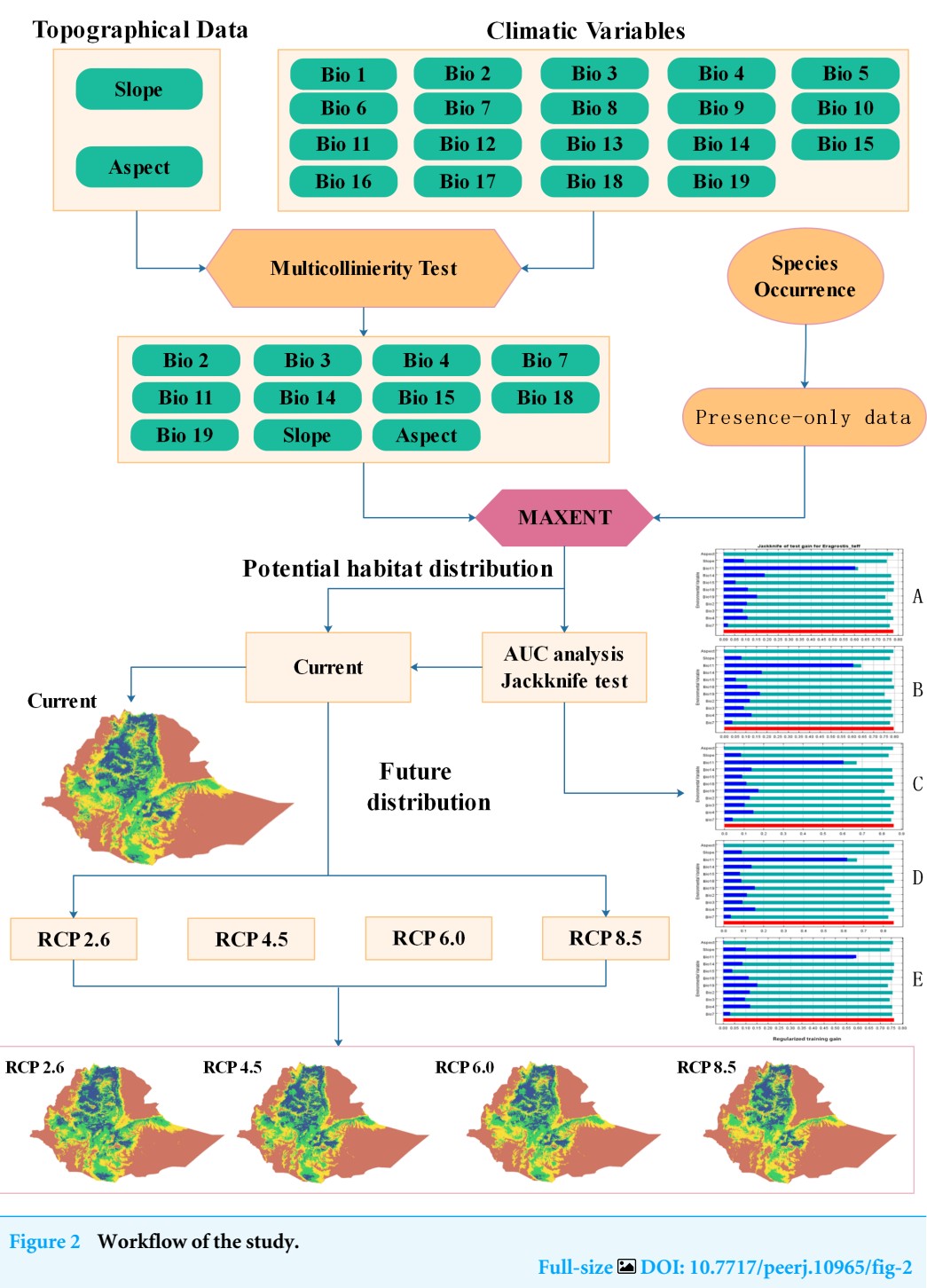

**Figure 2** Workflow of the study.

from desert climate to that typical of equatorial mountains (tropical) with three major climatological seasons: June–September (called Kiremt), October–January (Bega), and February–May (Belg) (*Gissila et al., 2004*; *Korecha & Barnston, 2007*; *Fazzini, Bisci & Billi, 2015*). Annual average temperature ranges from 10 to 27 ° C and precipitations vary across

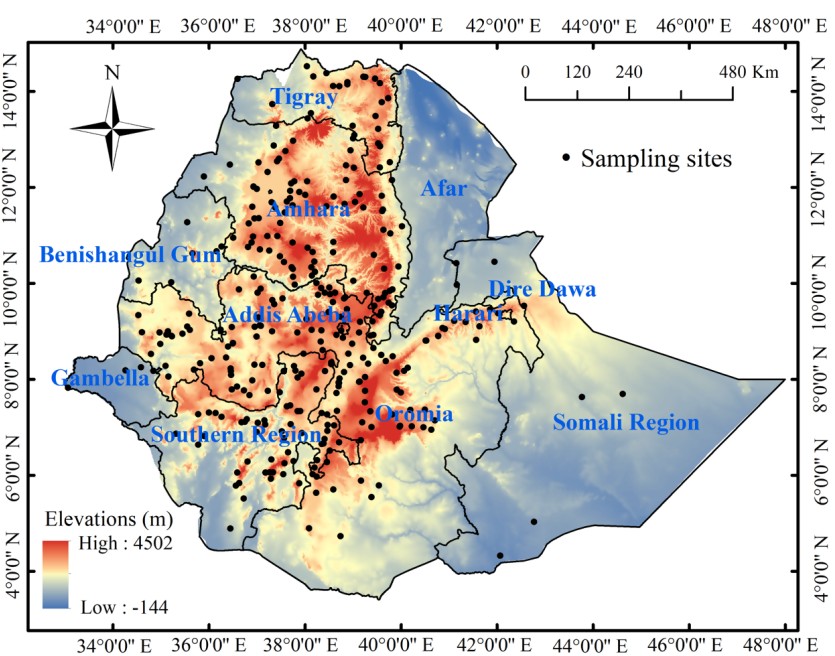

**Figure 3** Locations of sampling sites and land elevation within the nine Ethiopian regional state administrations: Oromia, Amhara, Tigray, Afar, Benishangul-Gamuz, Gambella, Harari, Southern Region, and Somali Region.

the country and season from 145 to 2,100 mm annually (*Hordofa et al., 2008*; *Fazzini, Bisci & Billi, 2015*).

## Crop production in Ethiopia

Though agricultural planning is difficult due to variable rainfall, a large proportion of Ethiopia receives sufficient rain for crop production, that is mostly sown from June to October and harvested from September to February (*Taffesse, Dorosh & Gemessa, 2012*). The primary crop season corresponds to the summer rainy season, from June to August and in the autumn from September to November (*Yumbya et al., 2014*). Light rains also fall during spring, from March through May. Some crops -only about 8% of total crop land- are harvested between March and August, making Ethiopia's crop season somewhat bimodal (*Hordofa et al., 2008*; *Taffesse, Dorosh & Gemessa, 2012*). Teff grows primarily in the highlands where clouds are forced to release rain. The optimal growing conditions for teff are 430–560 mm of rain per year and a temperature range of 10–30 ° C (*Roseberg et al., 2005*).

## Teff distribution data

The distribution dataset contains 2490 verified, geo-referenced data-points (latitude, longitude and altitude) belonging to germplasm collections and herbarium specimens from the gene bank of the Institute of Biodiversity Conservation, and 14 specimens from the Ethiopian National Herbarium, Addis Ababa University, collected between 1978 and 2019. Teff was present in all sites over the nine Ethiopian regional state administrations,

predominantly in the Oromia, Amhara, Tigray, the Southern Nations, and Harari regions, with only a few locations in Gambela, Beninshangul-Gumuz, and the Somali Region, and had no significant presence in Afar.

## Topographical data

The Digital Elevation Model (DEM) with a 90-meter special resolution was acquired from the USGS Shuttle Radar Topography Mission dataset (https://www.usgs.gov/centers/eros/science/usgs-eros-archive-digital-elevation-shuttle-radar-topography-mission-srtm-1-arc?qt-science_center_objects=0#qt-science_center_objects). Two terrain variables (aspect and slope) from the DEM were re-sampled into 1 km spatial resolution using neighbor sampling in ArcGIS 10.5. The two topographical variables were used as model inputs.

## Current and future climate data

Climatic data consisting of 19 bioclimatic variables, here coded Bio1 through Bio19, were obtained from the WorldClim (http://www.worldclim.org/) at 30 ′ ′ resolution (1 km ×1 km) for the climate from 1950 to 2000 in the study area (*Fick & Hijmans, 2017*). The dataset comprises global gridded information with 1 km spatial resolution on past, current, and future scenarios (http://www.worldclim.org/).

Future climate data is based on the standard scenarios of the Fifth Assessment Report from the IPCC (*IPCC, 2013*). All four greenhouse gas concentration trajectories provided by the IPCC were used, Representative Concentration Pathway (RCP): RCP 2.6, RCP 4.5, RCP 6.0, and RCP 8.0. In 2100, the total radiative forcing values in the four scenarios stated would have attained 2.6W/m2 ,4.5 W/m2, 6.0 W/m2, and 8.5 W/m2 over the value in the preindustrial period (*Taylor, Stouffer & Meehl, 2012*; *Rogelj, Meinshausen & Knutti, 2012*; *Stocker et al., 2013*). The data used was produced by interpolation of data predictions for 2060 and 2080 to produce an estimate of climate condition for the year 2070.

Cross correlation among 19 bioclimatic variables and two independent variables was checked to avoid the presence of redundant variables in the model. The input rate of the variables was calculated by Jackknife test of Maxent model, and IBM-SPSS statistical software (Statistical Product and Service Solutions, Version 20.0, SPSS Inc., Chicago, IL, USA) was used for Pearson's correlation. From each highly correlated pair of variables, one variable was removed (> |0.8|) and the ones with high contribution rate were chosen (*Liu et al., 2005*).

## Climatic niche modeling

We used Maxent Version 3.4.1 (https://biodiversityinformatics.amnh.org/open_source/maxent/; *Phillips, Anderson & Schapire, 2006*) to predict the current and future potential distribution of teff. Species distribution point data was stored in csv file format and included coordinate data such as name, latitude and longitude of the species. ArcGIS 10.5 converted all environmental variables to ASCII raster grids and species presence coordinates to decimal degrees. Certain settings (500 iterations, 0.00001 convergence requirement, 10,000 peak context points) were kept by default. The software ran with 75% of presence locations and tested the efficiency of the model with the remaining 25% of the presence points (*Phillips, Anderson & Schapire, 2006*; *Phillips & Dudík, 2008*). In
*Phillips, Anderson & Schapire (2006)*, a detailed mathematical description of Maxent was mentioned.

A continuous probabilistic layer ranging from 0 to 1 is the output of the model. Higher-value areas indicate more desirable conditions for growth of species (*Phillips, Anderson & Schapire, 2006*). For each case, we selected the minimum performance by training presence (MTP) as a threshold or "cut-off" value (*Hijmans & Graham, 2006*). The MTP can be interpreted ecologically to include those cells that are expected to be at least as sufficient as those where the species has been defined as present. We have categorized four habitat suitability groups, unsuitable, low suitability, moderate suitability, and high suitability by Spatial Analyst tool in ArcMap 10.5, so as to measure the niche centroid from present to future climate change scenarios.

There are two ratification processes in the Maxent model: area under the curve (AUC) and Jackknife. Area Under the Curve (AUC) of Receiver Operating Characteristic (ROC) curves was used to verify the suitability of teff estimated by the model. The AUC represents the probability that a randomly chosen teff occurrence exceeds that of randomly choosing an absence. The AUC values lie between 0.5 and 1.0 (perfect prediction), and can be divided into five categories: outstanding ($>$0.9), good (0.8–0.9), approved (0.7–0.8), poor (0.6–0.7) and invalid ($<$0.6) (Elith 2006, *Suwannatrai et al., 2017*).

Jackknife is part of Maxent model and it is calibrated with all permutations of the groups using occurrence points and background data from k $-$1 spatial groups and then evaluated with the withheld and alternate estimates of which variables are most important in the model (*Hu & Jiang, 2011*). Variables that significantly affected the model were identified and chosen based on percent contribution to the model.

## RESULTS

### Variable contribution analysis

After testing for independence between climatic variables, the ones with low predictive power or high similarities to other variables in prediction were identified. Eleven variables (11) were selected for the model according to the contribution rate of each variable (Table 1). The potential teff distribution under current projected climate change was mostly affected by the mean temperature of the coldest season (Bio11 78.3%) and precipitation seasonality (Bio15 8.3%) (Table 1). The potential teff distribution under the RCP scenarios was also influenced mainly by Bio11 (77.8%–80.4%), Bio15 (7.3%–9.04%). The Pearson's correlation coefficients are included in the (Table S1).

The jackknife test indicated that the bioclimatic variables Bio11, Bio14, Bio19, and Bio4 provided huge gains when they were used independently to estimate potential teff distribution under current conditions and under RCP scenarios 2.6 to 6.0 (Fig. 4). The potential distribution of the species under the RCP8.5 scenario was most strongly associated with Bio11 (gain, 0.59), Bio2 (gain, 0.12), Bio19 (gain, 0.15) and Bio4 (gain, 0.13), respectively (Fig. 4E).

The response curves indicate the relationship between environmental variables and habitat suitability and can provide information on teff ecological niche. The ranges of

**Table 1  Percent contributions of the variables to teff distribution in the MaxEnt model.**

| Code | Environmental factor | Unit | Percent Contribution | | | | |
|------|----------------------|------|---------|--------|--------|--------|--------|
| | | | Current | RCP2.6 | RCP4.5 | RCP6.0 | RCP8.5 |
| BIO1 | Annual mean temperature | °C | | | | | |
| **BIO2** | **Mean diurnal range (Monthly max. min. and mean temp.)** | °C | 0.23 | 0.6 | 0.2 | 0.5 | 0.40 |
| **BIO3** | **Isothermality (Bio2/Bio7 ×100)** | | 1.11 | 1.3 | 0.9 | 1.3 | 0.81 |
| **BIO4** | **Temperature seasonality (standard deviation ×100)** | | 1.33 | 0.8 | 0.5 | 0.45 | 0.44 |
| BIO5 | Maximum temperature of warmest month | °C | | | | | |
| BIO6 | Minimum temperature of coldest month | °C | | | | | |
| **BIO7** | **Temperature annual range (Bio5–Bio6)** | °C | 2.17 | 2.6 | 2.3 | 2.85 | 2.39 |
| BIO8 | Mean temperature of wettest quarter | °C | | | | | |
| BIO9 | Mean temperature of driest season | °C | | | | | |
| BIO10 | Mean temperature of warmest season | °C | | | | | |
| **BIO11** | **Mean temperature of coldest season** | °C | 78.3 | 79.7 | 80.4 | 78.5 | 77.8 |
| BIO12 | Annual precipitation | mm | | | | | |
| BIO13 | Precipitation of wettest period | mm | | | | | |
| **BIO14** | **Precipitation of driest period** | mm | 1.3 | 1.1 | 0.8 | 1.35 | 1.49 |
| **BIO15** | **Precipitation seasonality (CV)** | | 8.3 | 7.3 | 7.8 | 7.76 | 9.04 |
| BIO16 | Precipitation of wettest season | mm | – | – | – | – | – |
| BIO17 | Precipitation of driest season | mm | | | | | |
| **BIO18** | **Precipitation of warmest season** | mm | 0.2 | 0.2 | 0.1 | 0.14 | 0.03 |
| **BIO19** | **Precipitation of coldest season** | mm | 3.7 | 2.9 | 3.5 | 3.90 | 4.32 |
| **SLOP** | **Slope** | ° | 3.3 | 3.2 | 3.5 | 3.09 | 3.10 |
| **ASP** | **Aspect** | ° | 0.1 | 0.1 | 0.2 | 0.1 | 0.09 |

**Notes.**
The variables in bold were key variables selected by their contribution rates and multicollinearity test.
RCP, Representative Concentration Pathway.

suitability for environmental variables were identified by the threshold of normal suitable habitats. The response curves of 8 variables of teff habitat suitability are illustrated in Fig. 5, as is the suitable range for each variable. The highest suitability areas for teff have a mean temperature of the coldest season (Bio11) between 14–19 °C, an annual precipitation seasonality (Bio15) between 60% to 122%, precipitation of the coldest season (Bio19) between 8.5–110 mm, a slope between 0 to 3°, a temperature annual range (Bio7, Bio5-Bio6) from 11.8 to 26.6 °C, a temperature seasonality (standard deviation ×100, Bio4) from 1000–1550, precipitation of the driest period (Bio14) from 0.3 mm to 20 mm, an isothermality (Bio3) above 68 and an aspect from 355° to 0° (Fig. 5).

## Performance and accuracy of the Maxent model

Of the climatic variables, mean temperature of the coldest quarter played the greatest role in the model in both percent contribution and permutation importance, which reflects the effect of a variable determined by impact to the model when the variable is absent (Table 2). However, in the variable analysis, mean temperature of the coldest quarter was not independent of mean temperature of the warmest quarter, or mean temperature of the

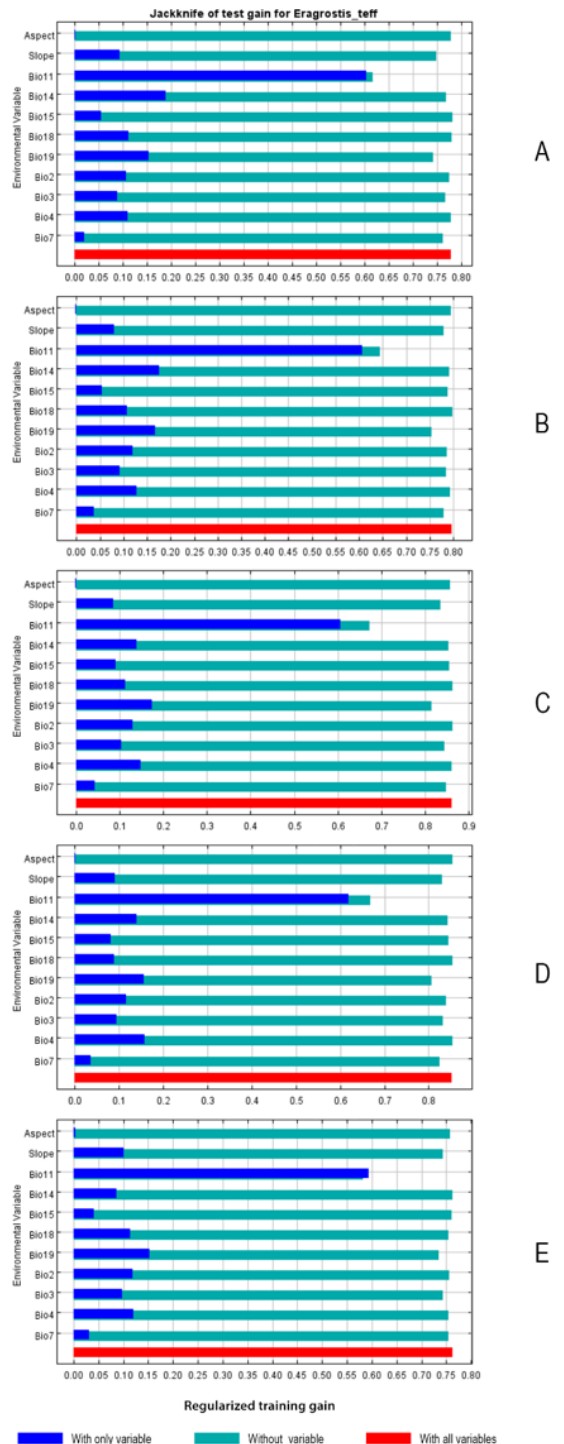

**Figure 4  Jackknife test variables contributions to potential distribution of teff distribution under (A) current climate condition scenario, (B) RCP 2.6 scenario, (C) RCP 4.5 scenario, (D) RCP 6.0 scenario, and (E) RCP 8.5 scenario.** The regularized training gain describes how much better the simulated distribution fits the present data compared to a uniform distribution. The dark blue bars indicate the gain from using each variable in isolation, the light blue bars indicate the gain lost by removing the single variable from the full model, and the red bar indicates the gain using all the variables.

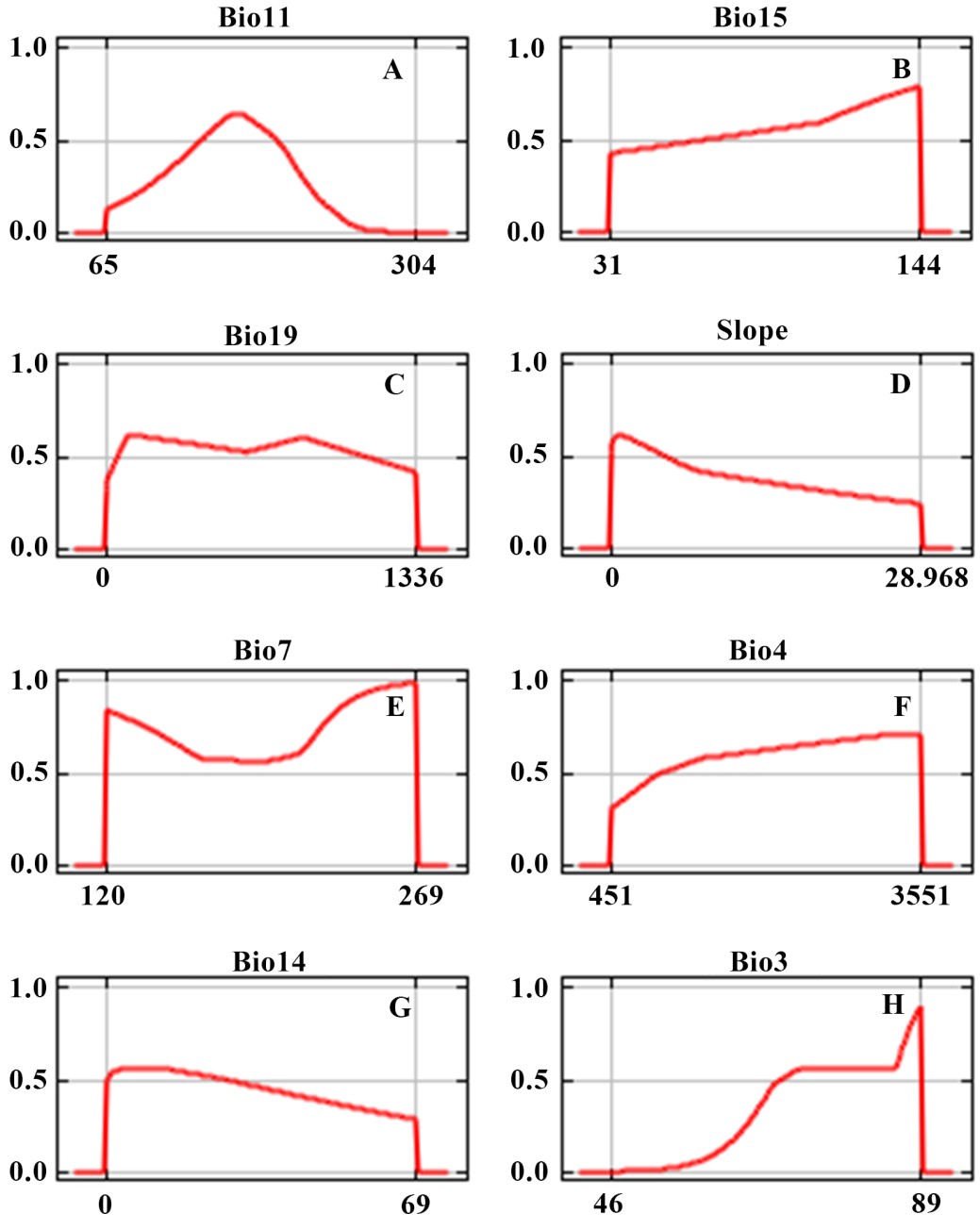

**Figure 5 Response curves of 8 environmental variables in the teff habitat distribution model.** (A) Bio11( Mean temperature of coldest season (° C*10)); (B) Bio15( Precipitation seasonality (CV)); (C) Bio7( Temperature annual range (Bio5–Bio6)); D: Bio19 (Precipitation of coldest season (mm*10)); (E) Bio14( Precipitation of driest period (mm*10)); (F) Bio3 (Isothermality (Bio2/Bio7 ×100)); (G) Bio4 (temperature seasonality (standard deviation ×100) (°C*10)); H (Slope).

driest quarter. Therefore, temperature generally had the greatest effect, especially during the growing season, which would logically be the highest for teff.

**Table 2  Estimates of relative contributions and permutation importance of the predictor environmental variables to the MaxEnt model.**

| Variable | Percent contribution | Permutation importance (%) |
|---|---|---|
| (Bio11) Mean temperature of coldest quarter | 78.3 | 66.5 |
| (Bio15) Precipitation seasonality | 8.3 | 4.7 |
| (Bio19) Precipitation of coldest quarter | 3.7 | 3.8 |
| Slope | 3.3 | 3.9 |
| (Bio7) Temperature annual range | 2.2 | 5.3 |
| (Bio4) Temperature seasonality | 1.3 | 3.1 |
| (Bio14) Precipitation of driest month | 1.3 | 1.8 |
| Bio3) Isothermality (mean diurnal range/temperature annual range) | 1.1 | 8.4 |
| (Bio2) Mean diurnal range (mean of monthly temp range) | 0.2 | 1.8 |
| (Bio18) Precipitation of warmest quarter | 0.2 | 1.8 |
| Aspect | 0.1 | 0.4 |

**Table 3  Results of receiver operating characteristic (ROC) analysis under current climate and four future projected scenarios.**

| AUC | Current | RCP2.6 | RCP4.5 | RCP6.0 | RCP8.5 |
|---|---|---|---|---|---|
| AUC (Training Data) | 0.841 | 0.839 | 0.835 | 0.835 | 0.843 |
| AUC (Test data) | 0.829 | 0.833 | 0.846 | 0.842 | 0.826 |

AUC values were above 0.829 under current and four projected climate scenarios (Table 3). According to the performance classification standard, prediction accuracy was very good, as AUC for training and testing were 0.84 and 0.83, respectively.

## Predicted distribution of teff under current climate condition and future scenarios

The model showed that teff has a climatically suitable area of minimum annual precipitation between 550 and 1m 770 mm. The minimum mean temperature is expected to be 14.9 °C with maximum values of 26.75 °C in Ethiopia. The model showed that under such temperature conditions, suitable distribution classes under the current and future scenarios showed that the warmer the climate condition, there will be a greater shift from high-level to low-level suitability.

The distribution classes under current and future scenarios show that teff in Ethiopia is highly suitable in the central region of the country. Areas with higher temperature that do not correlate with suitable teff habitat lie on the eastern side of highlands and those that do not correlate with higher temperature lie in the western lowlands (Fig. 6).

Generally, teff simulated distribution includes the west part of Ethiopia in current and future climate condition (Fig. 7). The high suitable area lies in the northwest part of the country, related to low temperature and high precipitation (Fig. 6). There is no distribution of teff in the eastern part of Ethiopia related to low altitude, high temperature and low precipitation (Fig. 6).
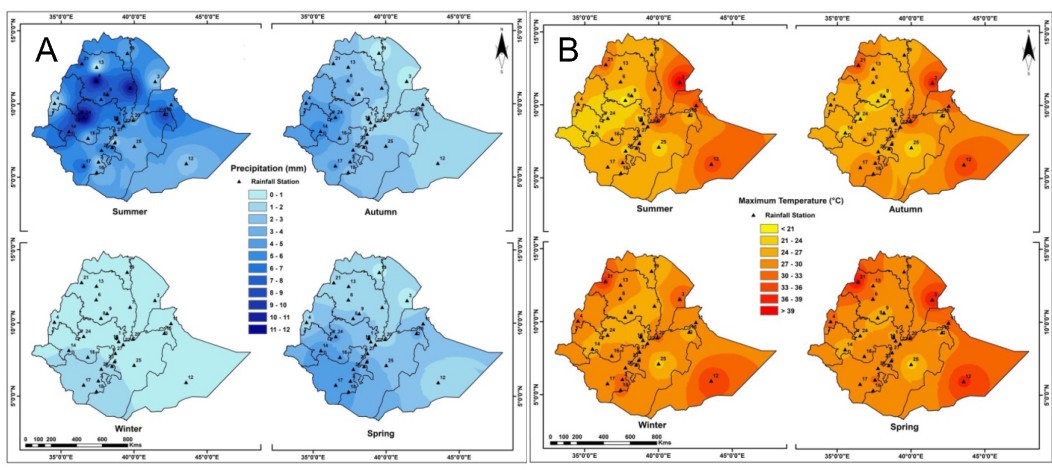

**Figure 6** Average annual rainfall (A) and average annual temperature (B) in Ethiopia over three decades.

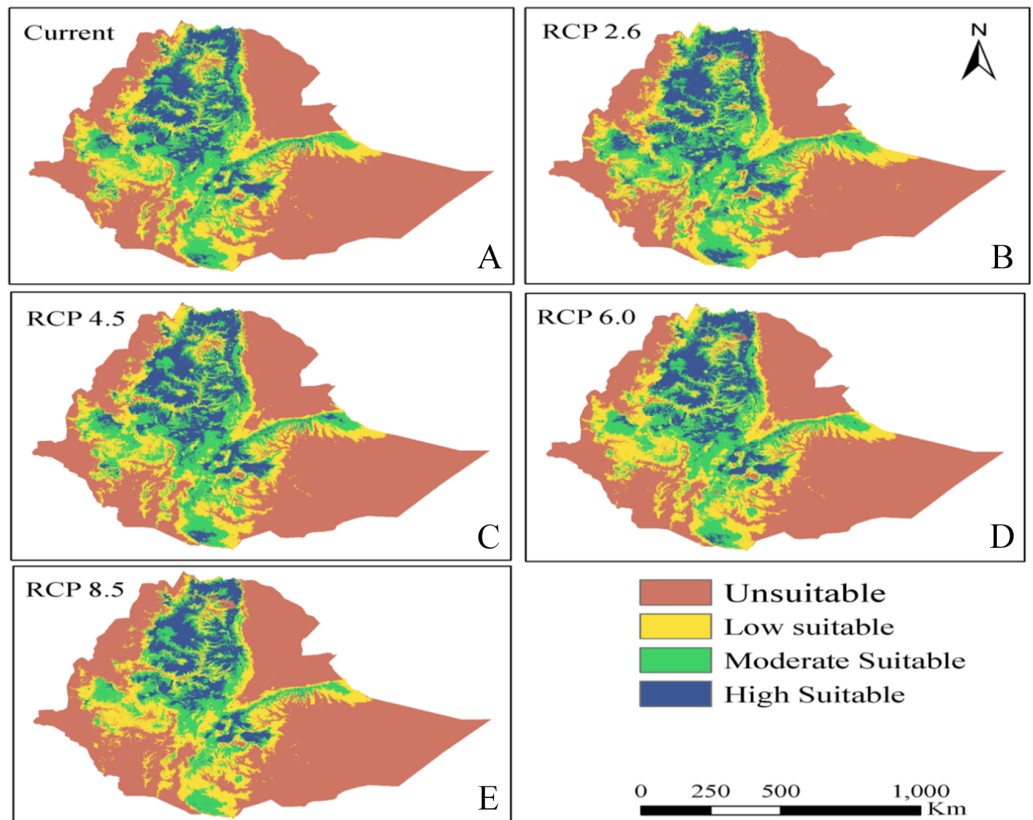

**Figure 7** Distribution of unsuitable, and low, moderate, and highly suitable teff habitats based on current and future distribution under four climate scenarios. (A) Current; (B) RCP 2.6; (C) RCP4.5; (D) RCP6.0; (E) RCP8.5.

**Table 4** The total area of suitable and unsuitable teff habitats based on current distribution data and projected four future climate scenarios.

|  | Current | RCP2.6 | RCP4.5 | RCP6.0 | RCP8.5 |
|---|---|---|---|---|---|
| Unsuitable area (km$^2$) | 560960 | 549902 | 565720 | 544774 | 568305 |
| Low suitability area (km$^2$) | 463453 | 463044 | 434560 | 464483 | 458548 |
| Moderate suitability area (km$^2$) | 296933 | 309811 | 320942 | 311603 | 293716 |
| High Suitability area (km$^2$) | 14513 | 13102 | 14637 | 14999 | 15290 |
| Total Suitability area (km$^2$) | 774899 | 785957 | 770139 | 791085 | 767554 |
| Change area (km$^2$) |  | 11058 | −4760 | 16186 | −7345 |
| Percent (%) | 58 | 58.8 | 57.6 | 59.2 | 57.4 |

## Teff habitat suitability under current and future scenarios

Under current climate, the distribution area with low suitability summed up 463,453 km$^2$, moderate suitability 296,933 km$^2$, and with high suitability, 14,513 km$^2$, giving a total suitable area of 58% of our research area. Considering the four projection scenarios, the total suitable area declined with climate warming in RCP 4.5 (770,139 km$^2$) and RCP 8.5 (767,554 km$^2$) while in other scenarios suitable area increased (Table 4, Fig. 7). A difference was found between the current suitable area and the predicted habitat in RCP 2.6 and 6.0 (Fig. 7). In particular, the area of suitable teff habitat increased in both RCP 2.6 and 6.0 while it decreased in RCP 4.5 (−4,760 km$^2$) and RCP 8.5 (−7,345 km$^2$) (Table 4).

Suitable habitat including moderately and highly suitable areas remained roughly the same in each future scenario (from 0.042 to 0.052 for RCP2.6 and 8.5 respectively). Taking the ratio of moderately suitable habitat to low suitable habitat, the largest projected deviation was under the RCP 4.5 scenario (ratio = 0.74 vs. 0.64–0.67 for the other RCPs), resulting in a relative decrease of low suitable area and an increase of moderate suitability. In the other scenarios, the model predicted a slight increase of highly suitable and a decrease of moderately suitable area (Table 4).

Unsuitable area in the future decreases in RCP 2.6 and RCP 6.0 compared to the current situation while it increases in RCP4.5 and RCP 8.5. Also, low suitability areas decrease in RCP 2.6, RCP 4.5 and RCP 8.5 while increasing in the RCP 6.0 scenario. Moderate suitability areas increase in RCP 2.6, RCP 4.5 and RCP 6.0 but decline in RCP 8.5. Again, high suitability areas compared with the current situation increase in area in RCP 4.5, RCP 6.0 and RCP 8.5 but not in the RCP 2.6 scenario (Table 4).

On the regional scale, the largest projected shifts in habitat suitability occur in the northwest region of the country under the most extreme radiative forcing case of RCP 8.5, where a patch of currently unsuitable area becomes moderately suitable (Fig. 7). Despite this single projected increase in area, overall, suitable land area is expected to decrease under RCP 4.5 and RCP 8.5, including a decrease in total moderately suitable land. The result of a shift in suitable land towards the west and away from the east contrasts with the path of rainclouds traveling from west to east.

## DISCUSSION

Ultimately, several studies and our own research results show it is necessary to assess the impact of climate change on teff distribution and take it into account in adaptive management policies in order to minimize the impact of climate change.

Recent research findings indicate that most updated climate projections of GCM (global climate model) used in the IPCC Fifth Assessment Report were used to forecast the effect of climate change on teff that was used for crop suitability (*Alemayehu et al., 2020*). The distribution of plants would be significantly affected by climate change according to the findings of several researchers. For example: *Mcsweeney et al. (2010)* and *Niang et al. (2014)*, reported that increased temperature and precipitation was projected in East Africa. This had an impact by decreasing the yield of many cereals by shortening the growing season length, amplifying water stress and increasing incidence of weed outbreak and diseases. As (*Prasad, Staggenborg & Ristic (2008)*; *Nelson et al., 2009* and *Thornton et al., 2009*) indicated that for high altitude regions in Ethiopia, such as mountainous land where temperature is the limiting factor for plant growth, a rise in temperature will possibly increase crop yield, but that in lowland areas, it will increase the risk of water stress. They also indicated that precipitation variability due to various effects of climate change, and the resulting heat and water stress, are considered the most important factors. *Elith & Graham (2009)* also reported that there is always uncertainty about selecting the best fit methods to model species distributions, as different models and methods change the result of the predictions under different scenarios due to various mechanisms. While there remains considerable uncertainty as to the extent and pace of global warming trends and their impact on plant species, preceding researchers have pointed out that climate change has a significant impact on the distributions of the different species by causing shifts, even contractions in species ranges, and change of terrestrial ecosystems function (*Barbet-Massin et al., 2012*; *Li et al., 2015*; *Petitpierre et al., 2016*). Considerable effort was made to expand the collection and increase the number of species studied. Due to limiting factors, such as climatic and environmental conditions, difficulty accessing some of the data and time constraints during occurrence sampling, some locations that were unreachable may have been overlooked. This shortcoming is believed to have slightly affected our results; hence, future studies should more comprehensively focus on the climate factor and other variables such as soil organic carbon, clay, bulk density (BD), Elevation, and pH. In future research, we should consider these methods to improve the accuracy of the predictors.

Ethiopia is predicted to be highly vulnerable to climate change, manifesting an increased warming accompanied by inconsistent rainfall (*Conway & Schipper, 2011*). In that direction, potential teff distribution under current conditions and projected climate change scenarios was mostly affected by the temperature and precipitation. *Evangelista, Young & Burnett (2013)* found that teff distribution depended most on precipitation variables, especially precipitation of the wettest quarter. However, we have found that the potential teff distribution was mostly affected by mean temperature of coldest quarter (Bio11), precipitation seasonality (Bio15) and precipitation of coldest quarter (Bio19).

As reported by *Tan et al. (2016)*, due to increasing temperature and varying rainfall the areas suitable for cultivating teff will become centralized to the plateau of Ethiopia in the future. Yet, (*Gregory, Ingram & Brklacich, 2005*; *Tognelli et al., 2009*) suggest that for most of the crops, including teff, suitability decreased due to the warming trend of climate change. Furthermore, *Thornton et al. (2009)* state that if temperature increases above the level suitable for a crop then it will lead to the death of that crop. Teff is known to be a primary cereal, which is less tolerant of cold (*Chamberlin & Schmidt, 2012*; *Yumbya et al., 2014*; *Sen et al., 2016*). Though our model share some common data and methods with preceding studies, it has arrived at different findings because it uses a wider range of data and environmental variables than those earlier studies. Existing research of different regions has reported variables found to have an important effect on various species. For example: *Evangelista, Young & Burnett, 2013* Studied teff by using three climate projection model that the value of predictable were 0.79 (Bio16, Bio12, Bio19, Bio15) and others studied different species in different geographical area including, (Elham et al., 2015) (Bio2, Bio3, Bio7, Bio8, Bio13, Bio14, Bio15, Bio18), (*Ardestani et al., 2015*) (Bio8, Bio19, Bio2, Bio13, Bio7), *Sen et al., 2016* (Bio4 and Bio19), *Rong et al., 2019* (Bio1, Bio8 and elevation), (*Abdelaal et al., 2020*) (elevation, precipitation, temperature and soil), and *Yang et al., 2020* (Temp Seasonality, Precipitation, Vegetation Type, Soil Type). Our results indicate that temperature of coldest quarter had a predictability value of 0.83 for teff, which is stronger than that of any other existing model. Mean temperature of coldest quarter (Bio11), Precipitation seasonality (Bio15), Precipitation of coldest quarter (Bio19), Slope, and Temperature annual range (Bio7) were the variables with the greatest effect on teff distribution.

Our result indicates that the suitable and unsuitable area somehow varies, increasing and decreasing under four climate change scenarios (i.e., RCP2.6, RCP4.5, RCP6.0 and RCP8.5). Comparing unsuitable area in current condition with that in scenario RCP 2.6 and RCP 6.0, unsuitable area decreased by 11,058 and 16,186 $km^2$ respectively. Also, the RCP 4.5, RCP 8.5 decreased by 4,760, and 7,345 $km^2$ respectively. Suitability area decreased comparing the current condition with RCP 8.5, suitability area increased comparing RCP 8.5 with the other three scenarios due to different climate changes under four scenarios. Future climate change scenarios RCP 2.6 and RCP 6.0 provided the most favorable outcomes in terms of loss of suitable teff habitat. However, RCP 2.6 resulted in a greater loss of high suitable habitat, whereas projections from RCP 6.0 show the highest preservation of total potential crop area. Surprisingly, projected loss of suitable habitat did not correlate with increasing projected radiative forcing. *Alemayehu et al. (2018)*, in relation to future climate change from 2050–2070s, indicated that generally suitable area for teff under climate change scenarios will suffer a slight increase in almost all administrative regions of Ethiopia. This increase in suitability will transform marginalized currently unsuitable areas to suitable area for teff under climate change scenarios. An example of these areas are Afar and Somali regional states which have low precipitation and high temperature conditions, and are predicted to have increased precipitation and higher temperatures. In contrast with our results in the four scenarios, they demonstrate the possible future expansion of the cultivation of teff in the east lowlands of Ethiopia. In 2050, the area expects a temperature

range between 15 °C and 27 °C, and a compensating increase in rainfall between 600 and 1,900 mm where teff would be climatically suitable (*Yumbya et al., 2014*). Several studies indicate that changes in rainfall, temperature and seasonality in Ethiopia directly affect rainfed agriculture (*Evangelista, Young & Burnett, 2013*; *Gebre et al., 2013*; *Asfaw et al., 2018*; *Alemayehu & Bewket, 2017*; *Worku et al., 2019*). *Evangelista, Young & Burnett (2013)* predicted a 350,000 km$^2$ loss of suitable crop area by 2050, however, in our study, areas of crop suitability showed different trends in the future climate change scenarios. In the model used by *Evangelista, Young & Burnett (2013)* rainfall was the primary climate driver (and not coupled with mean temperature of the coldest season), so the model outcomes were heavily dependent on the specific rainfall predictions of the climate data. The main difference in the models lies in the confidence of an environmental variable versus the prediction of a climatic variable.

Most studies using models to predict the climate change in Ethiopia revealed that important climate variables influence teff distribution. The contribution rates of precipitation of wettest quarter (Bio16), annual precipitation (Bio12), precipitation of the coldest quarter (Bio19), and precipitation seasonality (Bio15) are 26.6, 9.3, 8.3, and 6.1%, respectively (*Evangelista, Young & Burnett, 2013*). By contrast, our analysis results show that contribution rates of four environmental variables (Bio11, Bio15, Bio19, and slope) are 78.3, 8.3, 3.7, 3.3% respectively suggesting that precipitation in the wettest quarter is even more important than previously thought.

Our model provided reasonably strong AUC predictions on the geographic changes in teff suitability throughout Ethiopia under projected climate change. Several studies reveal that all models have associated assumptions and extrapolation modes in time or space, and are subject to violating assumptions of species distribution models (*Wiens et al., 2009*). We used a Maxent model to show current teff distribution and to predict future climate impacts on it. Since the model relies on predicted future climate scenarios, an amount of uncertainty is added. We should have considered including management, soil and other teff varieties which are grown in Ethiopia in our model analysis. We strongly believe that our research could be improved by including management of farming practice and preparation for teff harvesting components, but we believe that lack of soil and other data in our analysis did not significantly affect the results.

Climate change affects the teff crop both in current and future scenarios. Results indicate that the differing future distributions are not uniformly affected by predicted changes in climate. Moreover, the predicted future damage is so severe that the survival of the Ethiopian agricultural sector, will be at stake unless adaptive policies are implemented. As Ethiopia, many other East African countries are facing the erratic effects of climate change with the same projections (*Deressa & Hassan, 2009*; *Di Falco & Veronesi, 2013*).

## CONCLUSIONS

Our work presents an opportunity for the agricultural sector, modelers and policy makers to examine the effect of climate change on teff so as to minimize negative impacts. Our result revealed that four environmental factors have an important influence on teff distribution,

namely Mean temperature of coldest quarter (Bio11), Precipitation seasonality (Bio15), Precipitation of coldest quarter (Bio19), Slope, and Temperature annual range (Bio7). Furthermore, based on current and future climate future scenarios conditions, we found out that for teff distribution in western Ethiopia the high suitable area lies on the northwest part of the county. We discovered climate change is happening and influencing the teff distribution, with a total projected loss 5,000–7,300 $km^2$ of suitable habitat. While this change does not pose substantial threat on the short term, in the long term it may have a significant impact on the future availability to grow teff as staple crop in Ethiopia.

### Funding
This work was supported by the Priority research program of Chinese Academy of Science (XDA) (NO. 20100102). The funders had no role in study design, data collection and analysis, decision to publish, or preparation of the manuscript.

### Grant Disclosures
The following grant information was disclosed by the authors:
Priority research program of Chinese Academy of Science (XDA):  20100102.

### Competing Interests
The authors declare there are no competing interests.

### Author Contributions
- Dinka Zewudie conceived and designed the experiments, performed the experiments, prepared figures and/or tables, authored or reviewed drafts of the paper, and approved the final draft.
- Wenguang Ding conceived and designed the experiments, authored or reviewed drafts of the paper, guiding, and approved the final draft.
- Zhanlei Rong and Yapeng Chang analyzed the data, prepared figures and/or tables, and approved the final draft.
- Chuanyan Zhao conceived and designed the experiments, analyzed the data, prepared figures and/or tables, authored or reviewed drafts of the paper, guiding, Advising, and approved the final draft.

### Data Availability
  Occurrence data are available in the Supplementary Files.

### Supplemental Information
Supplemental information for this article can be found online at http://dx.doi.org/10.7717/peerj.10965#supplemental-information.

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
