# Peer review of "Spatiotemporal dynamics of habitat suitability for the Ethiopian staple crop, Eragrostis tef (teff), under changing climate"

_PeerJ, doi:10.7717/peerj.10965_

## Round 0.1 · original submission · Major Revisions

We have collected 2 reports, both request that you make revisions. Please revise your manuscript accordingly.

Reviewer 1 ·

Basic reporting

The agricultural systems in African countries are mostly rain-fed, and it is important to predict the crop yield in Africa. Therefore, the topic of the paper is interesting and the background of the study is well addressed. But the paper is poorly prepared. There is no workflow and the result, discussion, and conclusion sections are confusing. It is very difficult to understand what is new in this paper. Authors should read more academic papers and try to learn how to organize a scientific paper.

Experimental design

There is no workflow, and the method is not well described.

Validity of the findings

It seems there are some experiment results relates to the objectives in the result section, but there is no further analysis, and the conclusion section does not relate to the objectives listed in the introduction section.

Additional comments

Major concerns:
There is no workflow included in the paper. So that the experiment design is not clear. The authors should clearly show how they address the three objectives. I think the methodology section should be re-structured.
Line 253: Is Jackknifing a toolkit used for variable importance analysis? The authors should explain the tool kit in the method section.
Line 294: Is MaxEnt used to predict the distribution of teff? if yes, please analyze the teff distribution maps predicted by MaxEnt.
The discussion section should compare the result in the paper with other existing studies, show the advantages, limitations, and uncertainties of the paper.
line 416; In the conclusion section, I did not see anything related to the three objectives listed in the introduction section.

Minor concerns:
line 84: "have increased 0.85 ° from 1880 to 2012 0.85 ° C"
line 141-142: "genetic algorithm for the production of rules (GARP), BIOCLIM (the model widely fused tools to predict current and future species distribution of response)"
What are the number is the x-axis of Figure mean?

·

Basic reporting

The paper is generally very interesting with great results. However, the flow of thought in the paper is poor. The introduction, for example, is poorly structured with several editorial errors. The sentence on lines 119 – 121 (see the attached annotated PDF document) suggests the information on Teff is over but all of a sudden, the authors (by the sentence on lines 122 - 125) again bring in climate issues (which were discussed earlier). My suggestion is that the authors rearrange the introduction to ensure a smooth flow of thought. I would advise that the authors finish talking about climate issues and how they affect food production before introducing Teff. Afterward it not suitable to bring in climate again in the introduction because it disrupts the flow of thoughts.

Also, the sentences on Lines 192 – 200 (under the subsection "Teff Distribution Data") should be removed entirely from the manuscript or moved to other sections like STUDY AREA or INTRODUCTION where they may best fit.

I also noticed the same back and forth movement of thoughts in the discussion section.

Finally, I think the abstract is too long and needs revision.

Experimental design

Very well written and well explained. However, the authors should address the the following comment:

The parameters used for running the MaxEnt model have been provided but no explanation is given. Did these parameters provide the best model performance, for example? It is important to tell the reader. I think the 75%/25% training/test split is more or less a rule of thumb so explanation is not necessary.

Validity of the findings

no comment

Additional comments

A few more comments:

1. How different is the information on cultivated area of Teff (Lines 114 & 115) from “Teff crops occupy 988,638.5 square kilometers” (Lines 130)? They appear confusing to the reader and should be revised if both are needed.

2. Stewart & Getachew (1962) which was used to support the sentence "It is estimated that Injera, made from teff, provides up to two-thirds of the food consumed by Ethiopians" is too outdated. The situation may have changed. I suggest the authors use a more current reference.

3. Please be consistent with your measurement scale. If you want to use , m^2, km^2, etc. then convert other values from ha to metric scale as well.

4. Did the authors preform any statistical significance testing? I ask because of a sentence like “The potential teff distribution under current projected climate change was significantly affected…”

5. Some headings have all words capitalized while others do not. Ensure consistency.

6. Change bio1, bio2, … to Bio1, Bio2, … so that they are consistent with the rest of the text.

---

## Round 0.2 · Major Revisions

Please find additional comments from Reviewer 1 - please revise to address these comments and resubmit.

Reviewer 1 ·

Basic reporting

The paper has been improved significantly and I could better understand the work with the revised version. Then, I still have some concerns.

Experimental design

it's fine.

Validity of the findings

The result need more explanations, please see the Comment in the "General comment".

Additional comments

1. I suggest to use a figure to show the workflow of this paper, for example, it seems there are two steps for this paper (1. variables importance estimation and 2. teff distribution prediction based on multiple climate conditions), the workflow figure should show inputs and outputs of each step and then link these several steps together. This workflow figure could make it clear to the readers how the authors have designed their experiments. In addition, its new section should be added as the "workflow of the study" to describe the workflow figure.

2. Line 331~333 and 364~369,can you explain why your results are different from the existing studies? As they have concluded that precipitation in the wet season affected spatial of teff but your conclusion is the lowest temperature in the coldest season affects teff distribution the most, so can you any principle which could support your conclusion? I think this is important as the teff prediction result is highly dependent on the high contribution factors.

3. Line 386~403, this paragraph is not related to any experiment results, it seems it is better to provide this background information in the "Introduction section".

4. I suggest further discuss the possible effects of climate data uncertainty (particularly the uncertainty of future climate data) on your teff prediction results.

5. The conclusions section could be further improved. I suggest summarizing the conclusions as (1) conclusion of variables importance, (2) conclusion of predicted teff distributions, and (3) effects of climate on teff distributions based on your results.

---

## Round 0.3 · accepted · Accept

I'm happy to accept your paper for publication in PeerJ. I would like to congratulate you on your work and thank you for considering PeerJ for its publication.

Reviewer 1 ·

Basic reporting

All my concerns have been addressed and I am satisfied with the revision.

Experimental design

It's fine.

Validity of the findings

It's OK.

Additional comments

All my concerns have been addressed and I am satisfied with the revision.